# Plasmon Resonant Two-Photon Luminescence Inducing Photosensitization and Nonlinear Optical Microscopy In Vivo by Near-Infrared Excitation of Au Nanopeanuts

**Lun-Zhang Guo** [1,†]**, Cheng-Ham Wu** [1,†]**, Ming-Fong Tsai** [2,†]**, Fong-Yu Cheng** [3,†]**, Vijayakumar Shanmugam** [2]**, Zen-Uong Tsai** [1]**, Zhiming Zhang** [4]**, Tzu-Ming Liu** [4,*]** and Chen-Sheng Yeh** [2,*]

1   Department of Biomedical Engineering, National Taiwan University, Taipei 10617, Taiwan; lzguo0611@gmail.com (L.-Z.G.); st9001666666@gmail.com (C.-H.W.); tsaizu@gmail.com (Z.-U.T.)
2   Advanced Optoelectronic Technology Center, Center for Micro/Nano Science and Technology, Department of Chemistry, National Cheng Kung University, Tainan 701, Taiwan; l38982079@gmail.com (M.-F.T.); vijayakumarshanmugam@gmail.com (V.S.)
3   Department of Chemistry, Chinese Culture University, Taipei 111, Taiwan; zfy3@ulive.pccu.edu.tw
4   Ministry of Education Frontiers Science Center for Precision Oncology, Faculty of Health Sciences, Institute of Translational Medicine, University of Macau, Macau 999078, China; yc17601@connect.um.edu.mo
*   Correspondence: tmliu@um.edu.mo (T.-M.L.); csyeh@mail.ncku.edu.tw (C.-S.Y.)
†   L.-Z. Guo, C.-H. Wu, M.-F. Tsai, and F.-Y. Cheng contributes equally to this work.

**Abstract:** Photodynamic therapy (PDT) provides a potential therapeutic approach for killing malignant cell/solid tumors, but currently approved photosensitizers (PSs) are generally excited by visible light, limiting the penetration depth in tissues. It is necessary to develop a near-infrared (NIR) responsive photodynamic platform, providing maximum tissue penetration. Here, we present a gold nanopeanut platform exhibiting dual functions of NIR PDT and two-photon luminescence imaging. The nanopeanut with a size less than 100 nm exhibits two distinct NIR surface plasmon absorption bands at approximately 1110 and 1300 nm. To perform PDT, we conjugated commercial toluidine blue O (TBO) PS on the surface of the nanopeanuts. With spectral overlap, the 1230-nm femtosecond Cr: forsterite laser can excite the surface plasmons of nanopeanuts, transfer energy to TBO, and generate singlet oxygen to kill cells. Moreover, the plasmon resonance-enhanced two-photon luminescence of nanopeanuts can be used to map their delivery in vivo. These results demonstrate that the PS-conjugated gold nanopeanut is an effective theranostic system for NIR PDT.

**Keywords:** Au nanorod; nanopeanut; two-photon fluorescence; photodynamic therapy; fluorescence resonance energy transfer

## 1. Introduction

Photodynamic therapy (PDT) is a light-activated chemotherapeutic treatment that utilizes singlet oxygen and reactive oxygen species to react with surrounding biological substrates, such as mitochondria, lipid membranes, nucleic acids, and proteins. The oxidative reaction either kills or irreversibly damages malignant cells. The process requires light at the appropriate wavelength to activate the photosensitizer (PS) accumulated in diseased tissues. Then the PS enters excited triplet states and interacts with ground-state triplet oxygen, yielding cytotoxic species, e.g., excited-state singlet oxygen or free radicals. Unfortunately, visible activation wavelengths cannot penetrate tissues deeply and seriously impede this therapeutic strategy for currently approved PSs.

The step toward clinical practice prefers excitation with the least-invasive near-infrared (NIR) light, through which both blood and soft tissues have mild backscattering and low energy absorption, thus providing maximal tissue penetration. There are two approaches to generate singlet oxygen at the NIR wavelength. One primary method relies on chemical

synthesis to yield PS with a large two-photon absorption cross-section [1–4]. Another appealing strategy is taking fluorescence resonance energy transfer (FRET) as a tool to provide two-photon excitation of PS through the incorporation of a two-photon chromophore, e.g., dye molecules, acting as a donor to transfer energy for the excitation of a PS acceptor [5–8]. Since noble metal nanostructures have shown promising two-photon luminescence by their surface plasmon resonance (SPR), [9–14] the NIR-excited plasmons may serve as another kind of energy donor to excite PS. Herein, we present a platform exhibiting NIR PDT by taking advantage of the efficient nonlinear optical processes in core-shell gold nanopeanut structures [14]. The efficient two-photon SPR of gold nanopeanuts provides an energy source to excite PS for singlet oxygen generation. Importantly, the entire process can be triggered by the excitation of NIR wavelengths located in the second biological window (1000–1350 nm). Operating PDT at this wavelength regime offers better tissue penetration relative to 650–950 nm [15,16]. For example, hemoglobin has much less absorption in the NIR wavelength, and blood is sufficiently transparent from 1000 to 1350 nm. Notably, this nanopeanut particle (length: 53 nm, width: 22 nm) with a size less than 100 nm exhibits two distinct NIR plasmon bands at approximately 1110 and 1300 nm. We employed a 1230 nm femtosecond (~100 fs) Cr: forsterite laser to resonantly excite the surface plasmons and yield efficient two-photon luminescence of nanopeanuts. Meanwhile, commercial toluidine blue O (TBO) was conjugated on the surface of the nanopeanut. With a spectral overlap of SPR and TBO absorption bands at 600–650 nm, NIR-excited SPR can enhance two-photon absorption and transfer the excited charges to TBO PS to generate singlet oxygen (Figure 1). This new NIR PDT nanostructure also presents a nonlinear optical imaging capability to serve as a theranostic agent. Nonlinear optical imaging is known as an attractive technique for in vitro and in vivo biological applications because of its superior capability of deep-tissue observation. In this study, we found that the SPR effect resulted in improved two-photon fluorescence from TBO-conjugated nanopeanuts, giving stronger imaging contrast than nanopeanuts alone (no TBO conjugation) and making it a more promising theranostic agent in deep tissue.

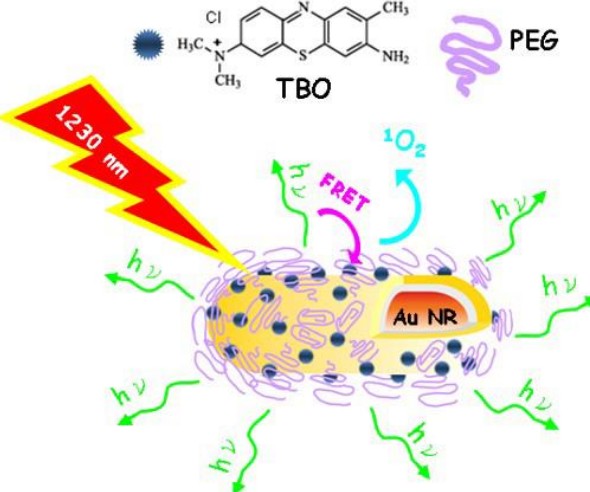

**Figure 1.** TBO-conjugated PEGylated nanopeanuts were activated by the 1230 nm femtosecond laser to generate singlet oxygen via fluorescence resonance energy transfer and yield two-photon luminescence.

## 2. Materials and Methods

### 2.1. Chemicals

All working solutions were prepared using doubly distilled water. Chemicals used were of analytical grade hydrogen tetrachloroaurate (III) hydrate (HAuCl$_4$ 3H$_2$O, Alfa Aesar, Haverhill, MA, USA, 99.99%), cetyltrimethylammonium bromide (CTAB, Fluka, Munich, Germany, 96%), ascorbic acid (AA, Riedel-deHaen, Seeize, Germany, MW 176.13, 99.7%), silver nitrate (AgNO$_3$, Sigma–Aldrich, St. Louis, MO, USA, MW 169.87, 99.8%), sodium borohydride (NaBH$_4$, Sigma–Aldrich, MW 37.83, 99%), polyvinylpyrrolidone (PVP, Sigma–

Aldrich, MW 130,000), 3-mercaptopropionic acid (MPA, Fluka, MW 106.14, 99%), toluidine blue O (TBO, Acros Organics, Antwerp, Belgium, MW 305.82), bis(amine)polyoxyethylene glycol (PEG, Sigma–Aldrich, Mw 106.14, 99%), toluidine blue O (TBO, Acros Organics, MW 305.82), bis(amine) polyethylene glycol (PEG, Sigma–Aldrich, MW 2100), and sodium carbohydrate (NHS, Sigma–Aldrich, MW 158).

### 2.2. Instrumentation

Low-magnification transmission electron microscopy (TEM) images were recorded using a Hitachi H-7500 electron microscope, and high-resolution transmission electron microscopy (HRTEM) images were recorded using a JEOL JEM-2100F electron microscope. Inductively coupled plasma (ICP) analysis for elemental composition was performed using a Jobin Yvon JY138 spectral analyzer. For the ICP analysis, nanostructures were digested in aqua regia. Absorption spectra of all the nanostructures were recorded in the UV-Vis-NIR region using a UV-Vis-NIR spectrophotometer (JASCO, V-670). Surface charge density was measured by the Zeta sizer Nanozs90. Fluorescence spectra of the Au nanopeanut conjugates were measured using a HORIBA Fluoromax-4 spectrometer.

### 2.3. Preparation of Au Nanorods (Au NRs)

The starting Au NRs were synthesized using a previously reported method [17] with slight modification at room temperature. Typically, 100 mL of the 0.5 mM $HAuCl_4$ precursor solution (prepared by mixing 5 mL of 5 mM $HAuCl_4$, 5 mmole CTAB, and making the volume to 50 mL with $H_2O$) was vigorously stirred, and 120 μL of the 100 mM $AgNO_3$ solution was added dropwise. After 5 min, 600 μL of the 100 mM ascorbic acid (AA) solution was added, followed by 40 μL of the 1.6 mM $NaBH_4$ solution, and the mixture was stirred for 150 min. The suspension obtained was centrifuged at 13,000 rpm for 12 min. The supernatant was discarded, and the precipitate was repeatedly washed with $H_2O$ under sonication followed by centrifugation as described above. Finally, the Au NRs were dispersed in $H_2O$ for the subsequent Au NR@Ag synthesis.

### 2.4. Preparation of Au NR@Ag

Au NR@Ag core-shell-type structures were synthesized using the method reported in the literature at room temperature [18]. Briefly, Au NRs were redispersed in 50 mM CTAB solution to obtain 1 mL of NR solution with an Au concentration of 50 ppm. This solution was then added to 5 mL of 1 wt.% PVP solution, followed by the addition of 0.95 mL of 1 mM $AgNO_3$, 0.125 mL of 100 mM AA, and 0.25 mL of 0.1 M NaOH solutions, sequentially. The mixture was stirred slowly for 10 min. The orange–red-colored solution thus obtained Au NR@Ag core-shell type NRs. The resulting Au NR@Ag colloids were centrifuged at 10,000 rpm for 10 min. The supernatant was discarded. The precipitate was washed with $H_2O$ under sonication, followed by centrifugation under similar conditions finally redispersed in $H_2O$.

### 2.5. Preparation of Au Nanopeanuts

The Ag nanoshell in the Au NR@Ag obtained from the above procedure was replaced with Au by slowly adding different volumes of the 1 mM $HAuCl_4$ aqueous solutions ($V_{HAuCl4}$ = 140, 280, 420, or 700 μL). Typically, 490 μL of 0.94 mM Au NR@Ag (based on Ag ion concentration) was dispersed in 5.75 mL $H_2O$, followed by the sequential addition of 500 μL of 100 mM CTAB and 125 μL of 100 mM AA. This mixture was placed under vigorous stirring conditions, and different volumes of $HAuCl_4$ were added. After 16 min, the reaction was stopped, and the products were purified by washing with a saturated NaCl solution, followed by repeated $H_2O$ washing via centrifugation at 10,000 rpm for 10 min and then redispersed in $H_2O$ for further characterization.

### 2.6. TBO Conjugation to Nanopeanuts

After the centrifugation-wash procedure, the nanopeanuts were redispersed in 1 mL H$_2$O and made up to 50 μg/mL. To 1 mL of the above nanopeanut, 67 μL of 7.5 mM 3-mercaptopropionic acid (MPA) was added and then sonicated for 2 h. Subsequently, the mixture solution was allowed to react overnight. The suspension obtained was centrifuged at 10,000 rpm for 10 min at room temperature. The supernatant was discarded, and the precipitates were repeatedly washed with H$_2$O under sonication, followed by centrifugation as described above. The resulting precipitates were dispersed in 1 mL of phosphate-buffered saline (PBS). This MPA-modified nanopeanut solution was mixed with 50 μL of 10 mM EDC/NHS and sonicated for 30 min at room temperature. Then, 130 μL of 30 μM toluidine blue O (TBO) was added to the EDC/NHS reaction solution, and the reaction was continued under sonication at room temperature for another 6 h. The resultants were centrifuged at 10,000 rpm for 10 min at room temperature, and the supernatants were discarded. The precipitates were redispersed and washed with H$_2$O under sonication, followed by centrifugation as above until the supernatants were free of unconjugated TBO, which was confirmed by measuring the supernatant absorbance at 630 nm using a UV-Vis spectrometer.

### 2.7. PEG Conjugation to TBO-Nanopeanuts

TBO-nanopeanuts with a concentration of 50 μg/mL were dispersed in a PBS buffer solution. Then, 50 μL of 10 mM EDC and 50 μL of 10 mM NHS were mixed and sonicated for 30 min at room temperature. Following the addition of 100 μL PEG (10 mM), the reaction was allowed to continue under sonication at room temperature for another 6 h. The resultants were centrifuged at 10,000 rpm for 10 min at room temperature, and the supernatants were removed. The remaining precipitates were repeatedly washed with H$_2$O by centrifugation as described above. To assess the amount of PEG bound to the Au nanopeanut surface, a fluorescence tag *viz.*, Rhodamine B isothiocyanate (RhBITC) molecule with N=C=S group conjugate on amine group of PEG was used. Then, 20 μL of 1 mM RhBITC was added to the above TBO-PEGylated nanopeanut conjugates at room temperature and reacted for 6 h. R6G-conjugated TBO-PEGylated nanopeanuts were collected following the centrifugation-wash process. The supernatants were then subjected to fluorescence detection. The luminescence emission at 552 nm of RhBITC was measured to find the concentration of RhBITC bound to TBO-PEGylated nanopeanuts with the corresponding calibration curve.

### 2.8. Cell Culture

The human lung carcinoma cell line (A549) was cultured in Dulbecco's modified Eagle's medium (DMEM, Cellgro) plus 10% fetal bovine serum (FBS, Gem Cell) at 37 °C under 5% CO$_2$. The cells were collected by trypsinization, placed onto a 10-cm tissue culture Petri dish, and allowed to grow for 3 days.

### 2.9. Cytotoxicity Assay (MTT)

Cells were seeded at a density of $5 \times 10^3$ per well in a 96-well culture plate and incubated overnight at 37 °C with 5% CO$_2$. Different concentrations of nanopeanuts were added to the medium and incubated for 24 h in the dark to check the cytotoxicity. Then, the culture medium with nanomaterial was removed and replaced with a new culture medium containing MTT reagents (10%) (Sigma) followed by incubation for 4 h at 37 °C to allow for the formation of formazan crystals. The crystals were dissolved in DMSO (200 μL) and incubated in the dark for another 10 min, followed by centrifugation at 4000 rpm for 10 min. The supernatant was transferred to a new ELISA plate, and the absorbance was measured at 540 nm wavelength with an ELISA reader.

For laser irradiation experiments, the material-treated cells were exposed to a 1230 nm femtosecond Cr: forsterite laser. A549 cells were incubated with nanopeanuts (20 μg/mL) for 8 h at 37 °C. After removing uninternalized colloids via PBS wash, A549 cells were

subjected to 3 min of laser exposure, followed by an additional 16 h incubation, and then subjected to MTT measurements.

### 2.10. Singlet Oxygen Detection at Different Times Measured under a 633 nm Diode Laser

A mixture of 1 mL of 20 µg/mL TBO-PEGylated nanopeanuts and 10 µL of 1 mM Singlet Oxygen Sensor Green Reagent (Invitrogen) (Ex/Em: 505/525 nm) was prepared and then exposed to a 633 nm diode laser (200 mW/cm$^2$, beam spot: 1 mm$^2$) in the dark. The singlet oxygen production is measured with the luminance intensity.

### 2.11. Darkfield and Fluorescence Images Monitor

Darkfield and fluorescence images were obtained using the 10× objective lens of the CytoViva microscope. The A549 cells were seeded at a density of $12 \times 10^3$ cells per well in 8-well chamber slides and incubated for 24 h. PEGylated- and TBO-PEGylated nanopeanuts were tested with and without a laser in addition to two controls, viz., the cell alone and the cell exposed to the laser. Three hundred microliters of 10 µg/mL nanopeanuts in the medium were prepared and added for the corresponding treatment. After 4 h of incubation at 37 °C, the cells were washed with PBS buffer twice and then replaced with the medium, followed by 3 min of exposure to a 633 nm diode laser (200 mW/cm$^2$ beam with a 1-mm$^2$ spot area) in the dark. Then, after 16 h of incubation at 37 °C, each well was washed twice with PBS. The cells were then fixed using 4% paraformaldehyde/PBS for 30 min at room temperature. The paraformaldehyde/PBS mixture was removed and washed twice with PBS. Three hundred microliters of Triton X-100 (0.2%) were added per well and incubated for 5 min, followed by washing as described above. Finally, the nuclei were stained with Hoechst (blue color), and the cytoskeleton was stained with Alexa 594 (red color) for CytoViva microscope analysis.

### 2.12. Singlet Oxygen Detection from PEGylated Nanopeanuts and TBO-PEGylated Nanopeanuts Excited by an Infrared Femtosecond Cr: Forsterite 1230 nm Laser

We prepared 100 µL of 20 µg/mL TBO-PEGylated nanopeanuts, added 10 µL of 1 mM Singlet Oxygen Sensor Green Reagent (Invitrogen) (Ex/Em: 505/525 nm) per well in a 96-well culture plate, and then exposed the plate to an infrared femtosecond Cr:forsterite 1230 nm laser (104 mW/cm$^2$) in the dark. The singlet oxygen production was measured with the luminance intensity of probes.

### 2.13. In Vitro Microscopic Imaging of Cells under Treatment

A549 cells were treated with 20 ppm PEGylated and TBO-PEGylated nanopeanuts separately to evaluate the photodynamic effect. Nanopeanuts were incubated with cells for at least 12 h.

### 2.14. Multiphoton Nonlinear Optical Microscope with a Microincubator

The imaging system is a femtosecond laser-based multiphoton nonlinear optical microscope with submicron 3D spatial resolution. The laser wavelength operates at approximately 1230 nm, which falls in the penetration window (1200–1300 nm) of most biological tissues. The penetration depth at 1230 nm is about 425 µm. Our home-build laser has a 25 nm full-width at half maximum in the power spectrum, 100 fs pulse-width, and a 110 MHz pulse repetition rate. The average power after the objective was 60 mW, corresponding to a 5454 W instantaneous peak power for each laser pulse. Compared with the commonly used Ti: sapphire laser (700–1000 nm), this wavelength does not two-photon excite the Soret band of many endogenous fluorophores in cells and tissues and thus has minor on-focus damage. With these advantages of in vivo imaging in deep tissues, it has been widely applied in developmental biology studies [19] and human clinical use [20,21], Because most autofluorescence is suppressed, optical contrast agents that can efficiently excite approximately 615 nm would have high contrast and benefit deep tissue imaging. The laser beam was XY-scanned by a scanning unit (FV300, Olympus, Tokyo,

Japan) cascaded with an inverted microscope (IX71, Olympus, Tokyo, Japan). The laser beam then transmitted a multiphoton dichroic beam splitter (edged at 665 nm) and focused by a water immersion objective (NA = 1.2, 60X, Olympus). The generated two-photon fluorescence (TPF), second harmonic generation (SHG), and third harmonic generation (THG) were epi-collected by the same objective. The TPF signals (>665 nm) transmit a dichroic beamsplitter and are detected by a photomultiplier tube (PMT) in the scanning unit. The SHG and THG signals were reflected and then separated by another dichroic beam splitter edged at 490 nm. They were detected separately by two other PMTs. All three channels of signals were reconstructed to $512 \times 512$ images with software in a computer with a 2-Hz frame rate.

To image live cells, we used a micro incubator on the microscope to create an environment with a temperature of 37 °C and 5% $CO_2$/95% air. The temperature of the thermostat (LAUDA Ecoline Staredition RE 204) was set at 50 °C to achieve 37 °C at the distal ends of the objective. The vapor reached the micro incubator through the duct and maintained the micro incubator at approximately 37 °C. The gas controller (OkO Lab, Pozzuoli, Italy) supplied 5% $CO_2$ continually and kept the outlet at 1 atm absolute pressure. The water immersion objective with 1.2 NA was heated by a dual temperature controller (TC-144, WARNER Instrument, Hamden, CT, USA), so that the contacting objective could be maintained at approximately 37 °C. The average power density after the objective is approximately $6.5 \times 10^6$ W/cm$^2$.

### 2.15. In Vivo Imaging

The animals used for in vivo microscopy observation were ICR mice. Female ICR mice aged 4 weeks were purchased from the Animal Center of National Taiwan University Hospital. All mice received humane care in compliance with the institution's guidelines for the maintenance and use of laboratory animals in research. All of the experimental protocols involving live animals were reviewed and approved by the Animal Experimentation Committee of National Taiwan University Hospital. All mice were acclimatized in plastic cages and taken care of by the animal center. The body weight was 25 g at the time of the experiment.

We used isoflurane inhalation anesthetics for the static observation of mice under an in vivo nonlinear microscope due to their effectiveness, lack of side effects, and rapid washout, especially in consecutive time-course imaging. Isoflurane inhalation supplies oxygen during whole anesthetic periods and is suitable for quick anesthetic recovery. Anesthesia was maintained with isoflurane vaporized at up to 4% concentrations in the induction phase at 0.8–1.5% during prolonged experimental observations. We monitored the anesthetized mouse's reflexes and vital signs (94–163 breaths/min, 325–780 beats/min, 37.5 °C) and maintained the body temperature with a small warm bag during the whole anesthetic period throughout recovery.

## 3. Results

### Characterization of Au Nanopeanuts

Au nanorods (Au NRs) with an aspect ratio of ~4 (length: ~40 nm, width: ~10 nm) were first synthesized using the seedless method (Figure 2a) [17]. Following centrifugation and wash processes, Au NRs were collected and redispersed in $H_2O$ and then subjected to the formation of Ag nanolayers on their surface by the reduction of $AgNO_3$ in a mixture of ascorbic acid (AA), cetyltrimethylammonium bromide (CTAB), and polyvinylpyrrolidone (PVP) to yield Au NR@Ag [18]. The TEM image (Figure 2b) clearly shows differential contrast with darkened Au nanorods embedded in the Ag nanolayer. The thickness of the Ag nanolayer could be tuned relative to the change in $AgNO_3$ amount, and 7–8 nm Ag nanoshells were prepared in this study. Au NR@Ag products were collected and subjected to centrifugation and wash procedures for further nanopeanut formation. Au NR@Ag was converted to a nanopeanut structure with Au/Ag nanoshells in a reaction mixture containing $HAuCl_4$ (700 µL), CTAB, and AA rather than by pure addition of $HAuCl_4$

alone, which is a common approach for galvanic exchange. After a short reaction period of 16 min, the nanopeanut formed with an Au NR embedded in a hollow Ag/Au shell (Figure 2c). Typically, nanopeanuts have a size of ~53 nm in length and ~22 nm in width with ~6 nm in the thickness of Au/Ag nanoshells. We estimated the gap between Au NRs and Au/Ag nanoshells as 2 nm from TEM images (Figure S1). An energy dispersive X-ray analysis (EDX) of a single nanopeanut gives an average atomic ratio of Au/Ag of 47.5/52.5 in the shell (Figure S1). The most interesting observation is the evolution of UV-Vis-NIR absorption behavior in the course of preparation. Initially, Au NRs display transverse and longitudinal absorbance maxima at 521 and 800 nm, respectively (Figure 2a). Once the gold nanorod was epitaxially coated with an 8 nm silver shell, the transverse and longitudinal maxima blueshifted to 393 nm and 606 nm (Figure 2b), respectively, due to the different dielectric functions of Ag and Au [18,22]. In addition, a small shoulder appeared at 360 nm, which could be attributed to the anisotropic growth mode [23], indicating that Ag nanoshells were not evenly deposited on Au NRs. Upon replacement with gold, this thin silver shell grew into a rough dented Au/Ag shell around the core rod with a distinct dielectric gap between them. The optical behavior of nanopeanuts exhibits wide absorbance scaling from UV wavelength to NIR 1350 nm. As demonstrated in our previous work [14], this broadened absorption is due to the gap structure enhanced surface plasmon resonance. Interestingly, two separate bands stand up at 1110 and 1300 nm. The colloidal solutions changed from reddish (Au NRs) to grayish blue (nanopeanuts). The as-prepared Au NRs, Au NR@Ag, and nanopeanuts were all washed with $H_2O$ and centrifuged, and then redispersed in $H_2O$.

The absorbance bands appearing at NIR have driven us to tune the gap between Au NRs and Au/Ag nanoshells. Our preliminary examination showed that the additional amount of $HAuCl_4$ tunes the gap distance in the replacement reaction. Figures S2 and S3 show the UV-Vis-NIR, colloidal color, TEM, HRTEM, and EDX analysis of nanopeanuts prepared with 140, 280, and 420 μL in addition to 700 μL of $HAuCl_4$ for replacement reactions. For 140 μL $HAuCl_4$, we hardly observed the gap appearance, but the creation of the pinhole could be recognized (Figure S3). Increasing the $HAuCl_4$ volume to 280 μL resulted in a distinct dielectric gap of ~2.6 nm with a shell thickness of ~6 nm. When the gold ion volume was further increased to 420 μL, the gap increased to ~3.8 nm, whereas the shell thickness decreased to ~3.7 nm. Consequently, the change in shell width and gap tailored the UV-Vis-NIR behavior. The resulting colloids prepared using 140 and 280 μL $HAuCl_4$ display stronger absorbance below 1000 nm, while a reverse appearance for wavelengths above 1000 nm shows greater amplitude with 420 and 700 μL additions (Figure S2b). Based on these observations in the change of gap and shell thickness, we speculate that the coupling of the central Au NR and Au/Ag nanoshell seems to strongly affect those two NIR bands. However, we should also be careful to consider the composition in Au/Ag nanoshells. The EDX analysis indicates that the ratios of Au/Ag shells were 11.9/88.1, 30.6/69.4, and 34.3/65.7 when 140, 280, and 420 μL of $HAuCl_4$ was used, respectively. To gain insight into the UV-Vis-NIR spectral tunability behavior, it is necessary to carry out theoretical calculations as well as more detailed experiments with systematic control of the gap relative to Au/Ag composition. For example, Halas and coworkers developed a hybridization model to interpret plasmon modes for spherically concentric nanoparticles composed of Au nanoparticles surrounded by a silica layer and then an outer Au shell [24,25]. The plasmon resonance arises from the Au core to interact with the bonding and antibonding modes of the surrounding nanoshell. The final note for nanopeanut formation is that an additional feature was seen in the observation of the backfilling phenomenon from 420 to 700 μL. Figure S2 clearly shows the decrease in the gap and increase in the shell when subjected to 700 μL of $HAuCl_4$ addition. We found that both CTAB and AA in solution are essential to cause backfilling. With the current method of preparing nanopeanuts under a reaction solution containing $HAuCl_4$, CTAB, and AA, we could either fix the gap by changing the shell thickness or adjust the gap width

by maintaining the shell thickness. The destruction of morphology normally occurs in the replacement reaction when a high $HAuCl_4$ amount is used.

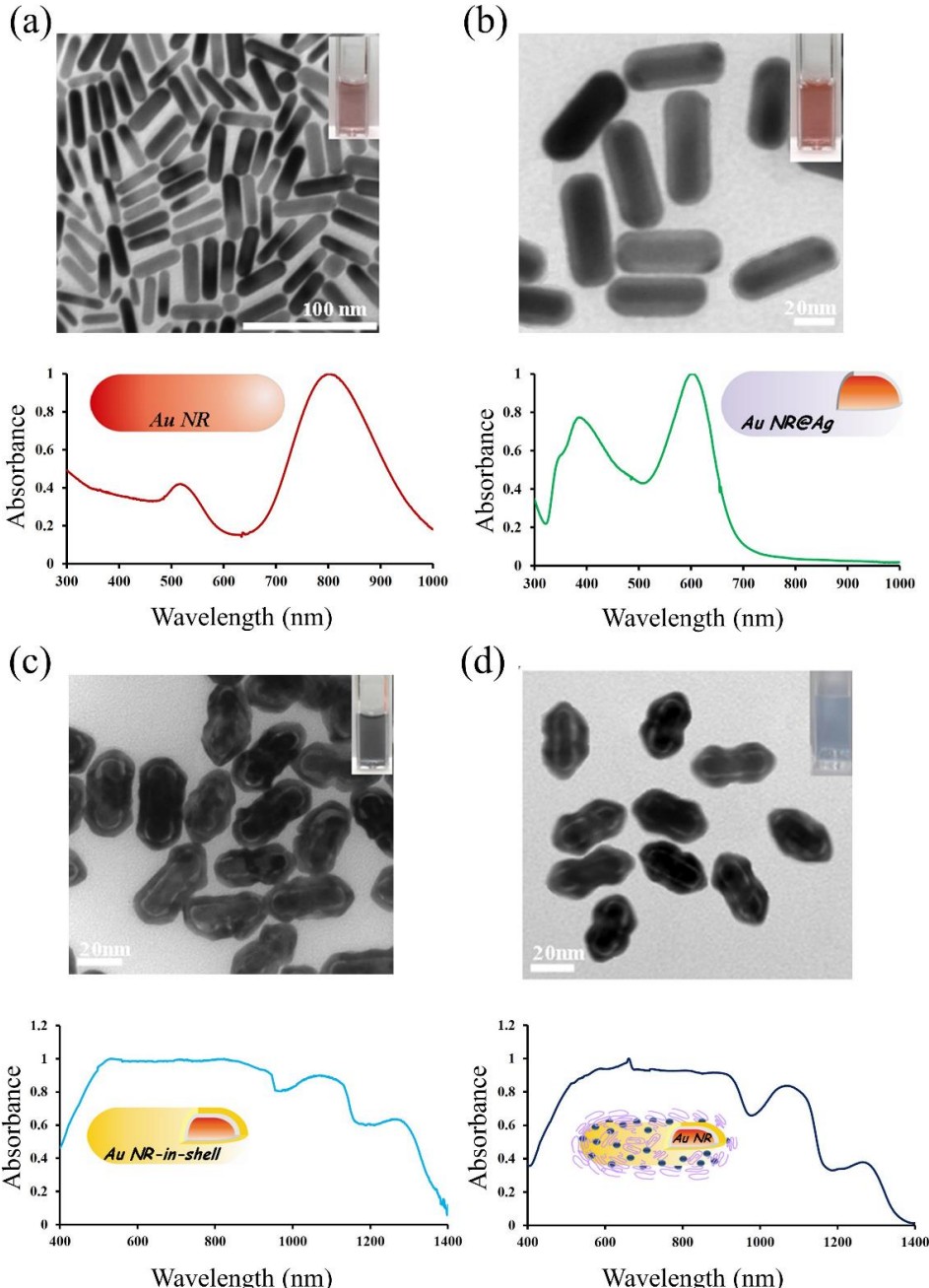

**Figure 2.** TEM images, colloidal aqueous solutions (inset), and UV-Vis-NIR absorption spectra of the resulting (**a**) Au NRs, (**b**) Au NR@Ag, (**c**) Au nanopeanuts, and (**d**) TBO-conjugated PEGylated Au nanopeanuts.

To perform two-photon photosensitization and two-photon luminescence imaging, the surface of the nanopeanuts, derived from the addition of 700 μL of $HAuCl_4$, was conjugated with TBO and bis(amine)polyoxyethylene glycol ($H_2N$-PEG-$NH_2$, MW: 6000). After the centrifugation-wash process, as-prepared nanopeanuts were collected and redispersed in $H_2O$ for surface modification. First, 2-mercaptopropic acid (MPA) was used to form an Au-S bond on the surface of nanopeanuts, resulting in the surface charge of nanopeanuts changing from 3.3 to −33 mV. Subsequently, the amine groups of TBO formed amide bonds with carboxylate groups of MPA through (1-ethyl-3-(3-dimethylaminopropyl) carbodiimide hydrochloride/N-hydroxysuccinimide (EDC/NHS) chemistry. The amount of immobi-

lization was measured from the decrease in absorbance intensity at 630 nm of TBO left in supernatants and calculated based on a calibration curve according to TBO concentration (Figure S4). The ~507 TBO was estimated to be on a single nanopeanut. The nanopeanut surface was not fully covered by TBO but rather had residual space left to tether hydrophilic amine-terminated PEG. Once again, amide bonds formed between MPA and PEG through the EDC/NHS reaction. The quantification of immobilized PEG was performed with a fluorescence-based method by the reaction of rhodamine 6G (R6G)-isothiocyanate with the exposed $NH_2$ group of PEG. The PEG conjugation amount was determined to be ~2432 molecules per nanopeanut, which was derived from the difference in fluorescence intensity of R6G between the initial amount and final residue in supernatants, calculated based on a calibration curve according to the R6G concentration. The resulting TBO-conjugated PEGylated (TBO-PEGylated) nanopeanuts dispersed in $H_2O$ lead to a light blue colloidal solution, which exhibits the same UV-Vis-NIR absorption spectral contour as the absorption spectral contour of nanopeanuts alone (Figure 2d). Interestingly, an additional peak appears at ~660 nm, a slight redshift from the absorption band (630 nm) of pure TBO. We think that this additional peak may arise from the interaction of the surface plasmon of nanopeanuts with TBO molecules [26,27].

Then we evaluated the cytotoxicity of TBO-PEGylated nanopeanuts (Figure 3). A series of particle concentrations were incubated with A549 cells in 96-well plates for 24 h and subjected to MTT measurements. The nanopeanut alone has shown a certain degree of toxicity at a concentration larger than 10 μg/mL, which is likely due to the Ag composition in nanoshells. Fourier transform infrared (FTIR) spectroscopy measurements show no CTAB features on the nanopeanut surface. In contrast, neither PEGylated nor TBO-PEGylated nanopeanuts displayed toxic effects at concentrations up to 100 μg/mL. Before performing two-photon activation experiments, we employed a 633 nm diode laser to verify that TBO-PEGylated nanopeanuts can generate sufficient singlet oxygen to cause photodynamic damage in malignant cells. Since the absorption band of TBO is located at 620–670 nm, a 633-nm diode laser can directly excite TBO (one-photon excitation) to yield singlet oxygen. With the Singlet Oxygen Sensor Green Reagent (Ex/Em: 505/525 nm) (Invitrogen), the direct detection of singlet oxygen was performed using a fluorescence spectrophotometer. Singlet Oxygen Sensor Green is a nonfluorescent molecule, but it can react with singlet oxygen to become a fluorescent one. The fluorescence intensity of Singlet Oxygen Sensor Green is proportional to the length of irradiation time, indicating that a longer irradiation time produces more singlet oxygen (Figure S5).

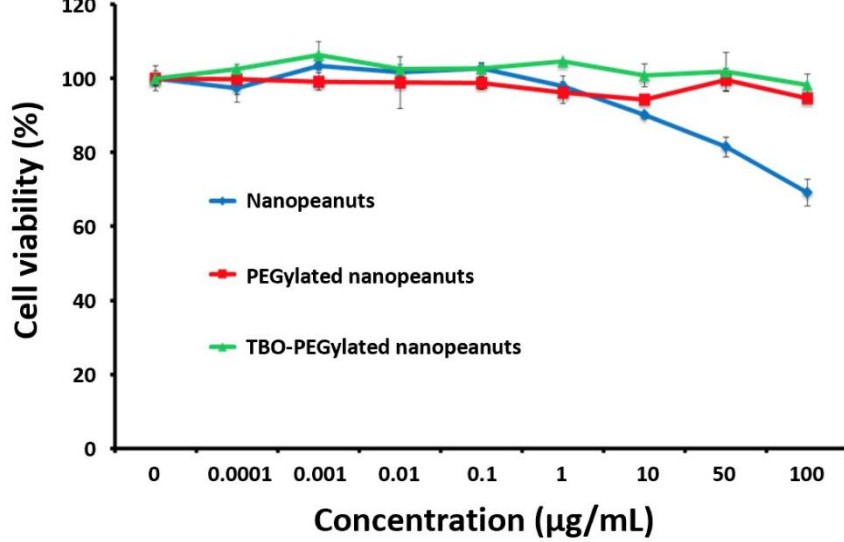

**Figure 3.** Cell viability studies of A549 cells incubated individually with different concentrations of nanopeanuts at 37 °C for 24 h.

Then, we used a dark field and fluorescence imaging system (CytoViva microscope) to monitor cell viability under illumination with a 633-nm diode laser (Figure 4). Laser irradiation at 633 nm had no effect on cell viability. After treating cells with TBO-PEGylated nanopeanuts (10 μg/mL) for 4 h at 37 °C, we observed cell uptake under darkfield scattering images. Without laser irradiation, it also showed no damage in cells. After removing uninternalized colloids by PBS wash, A549 cells were subjected to 3 min of laser exposure (200 mW/cm$^2$), followed by an additional 16 h incubation. Significant cell destruction was observed in the darkfield image. The disintegration of cytoplasm and the condensation of cell nuclei appeared in fluorescent images as well. These results indicate effective singlet oxygen generation and cancer cell damage upon 633-nm diode laser exposure. The low laser power (200 mW/cm$^2$) did not cause a temperature elevation of the colloidal solution, remaining at 25 °C after 20 min irradiation. There was no hyperthermic effect in the course of irradiation.

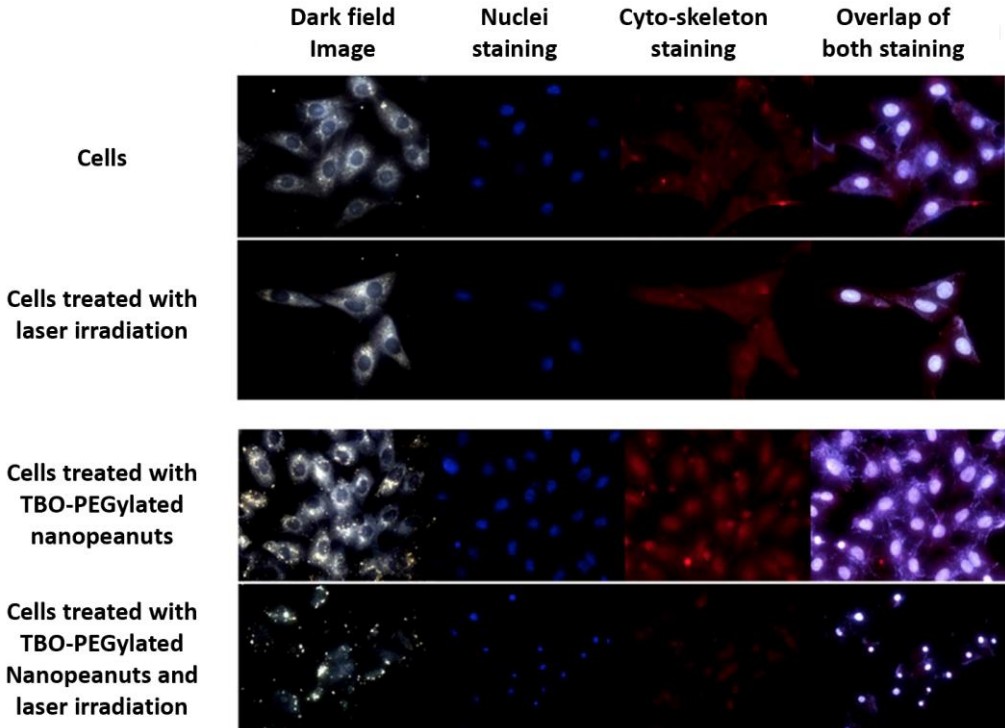

**Figure 4.** Darkfield scattering and fluorescence images of A549 cells exposed to either nil or 633-nm diode laser irradiation. A549 cells were incubated with TBO-PEGylated nanopeanuts (10 μg/mL) for 4 h at 37 °C. After removing uninternalized colloids by PBS wash, A549 cells were subjected to 3 min of exposure to a 633-nm diode laser (200 mW/cm$^2$), followed by an additional 16 h incubation. The images were acquired using a 10× objective lens (CytoViva microscope).

After characterizing the PDT capability of the TBO-PEGylated nanopeanut, we demonstrated its nonlinear optical property using a 1230-nm femtosecond Cr: forsterite laser. The multiphoton fluorescence spectra of Au nanopeanuts are broad and enhanced by the surface-plasmon resonance effect in the nanopeanuts (Figure 5, red curve). An additional band was observed at ~670 nm when TBO-PEGylated nanopeanuts were excited (Figure 5, black curve). Such TBO fluorescence was also observed when the TBO-PEGylated nanopeanuts were excited by a xenon arc lamp from a conventional fluorescence spectrometer (Figure S6). The resonantly enhanced surface plasmon field of the metal nanostructure is known to be able to interact with the transitional states of the nanomaterials, thereby resulting in enhanced luminescence intensity and the efficiency of excitation [28–30]. In the SPR-enhanced nonlinear optical processes, not only the local excitation electric field, but also the nonlinear polarization is enhanced. For nonlinear optical processes such as two-photon fluorescence (TPF), harmonic generation, and Raman scattering [31], the enhancement of

overall signal yields could be more than 100 [9,31–34]. In our gold nanopeanuts, the local field enhancement factor of surface plasmon resonance at 1230 nm is about 9 [14], which benefits the yield of two-photon luminescence and the singlet oxygen generation. Upon exciting 1230-nm femtosecond laser pulses for 3 min, the TBO-PEGylated nanopeanuts showed significant singlet oxygen generation, while no singlet oxygen was detected using PEGylated nanopeanuts (Figure S7). Then, we conducted NIR PDT in vitro in A549 malignant cells using the MTT assay (Figure 6). The photodynamic efficacy was evaluated using two different beam size conditions: A focused beam ($5.7 \times 10^5$ W/cm$^2$; beam diameter: 1.5 μm) and a collimated beam (104 mW/cm$^2$; beam diameter: 3–4 mm). Without laser exposure, cell viability was not affected by either PEGylated or TBO-PEGylated nanopeanut incubation. Additionally, no decrease in cell viability was observed for cells alone or for PEGylated nanopeanut-treated cells irradiated by NIR femtosecond laser. There are no photothermal damages on cells. For cells treated with TBO-PEGylated nanopeanuts, apparent cell death was observed from focused and collimated beam irradiation, giving 63% and 45% (<IC$_{50}$) viability, respectively. Higher cell death in the collimated beam illumination can be attributed to a larger exposure area and more production of singlet oxygen. These results verify the PDT capability of TBO-PEGylated nanopeanuts in the second biological window.

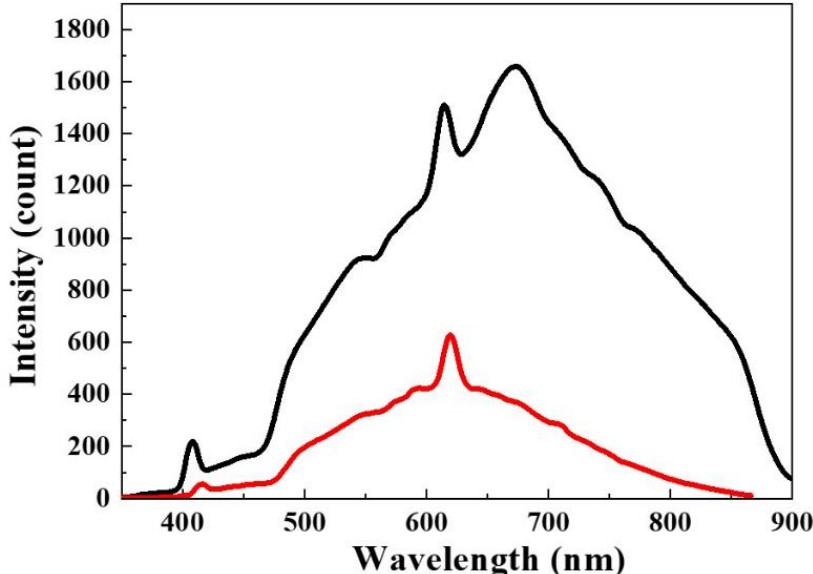

**Figure 5.** Multiphoton emission spectra of nanopeanuts alone (no surface modification, red curve) and TBO-PEGylated nanopeanuts (black curve). The emission peaks at approximately 410 nm and 615 nm are third-harmonic generation (THG) and second-harmonic generation (SHG) of nanopeanuts. The arising band at approximately 670 nm is the fluorescence from TBO.

Subsequently, we employed a multiphoton nonlinear optical microscope to observe microscopic evidence of PDT action. The cells were cultured and attached well to the bottom of glass Petri dishes. On a laser-scanned inverted microscope, these dishes were mounted within a micro incubator to keep cells in their normal physiological conditions. The femtosecond Cr: forsterite laser was used as the excitation source of the nonlinear microscope. Its minimal invasiveness for nonlinear microscopy imaging has been demonstrated in zebrafish embryos [31], mouse embryos, and human skin [32,33]. Without any treatment of nanopeanuts, the morphology of attached cells can be observed through a third harmonic generation (THG) modality without labeling (Figure 7a). Due to the 1230 nm excitation wavelength of the femtosecond Cr: forsterite laser, most endogenous fluorophores would not be resonantly excited and show no fluorescence background (Figure 7a), even with the highest photomultiplier gain. Long-term observation for 30 min did not change the attachments and morphologies of cells. For cells treated with PEGylated

or TBO-PEGylated nanopeanuts, apparent two-photon fluorescence (TPF, yellow color) contrasts can be observed (Figure 7b,c) within cells, indicating that the cells took up these nanopeanuts. As mentioned above, the TPF spectrum (Figure 5, black curve) of TBO-PEGylated nanopeanuts shows an extra contribution (~670 nm) from TBO, which could be the reason that TBO-PEGylated nanopeanuts (Figure 7c) show stronger TPF contrast than PEGylated nanopeanuts alone (Figure 7b). We surveyed a power level below the regime of photothermal effects while maintaining large enough photodynamic effects. As shown in Figure 8a, after 5 min of illumination, no apparent photodamage was done to the PEGylated nanopeanut-treated cells. No effect on cell morphology was seen even when the irradiation time was prolonged up to 15 min. In contrast, cells with the TBO-PEGylated nanopeanuts effectively shrank after 3 min of illumination (Figure 8b). These effects of the NIR laser on cells agree well with the results of singlet oxygen tests and MTT assays. They indicate that under 1230 nm femtosecond laser illumination, TBO-PEGylated nanopeanuts can produce singlet oxygen and cause cell death quickly.

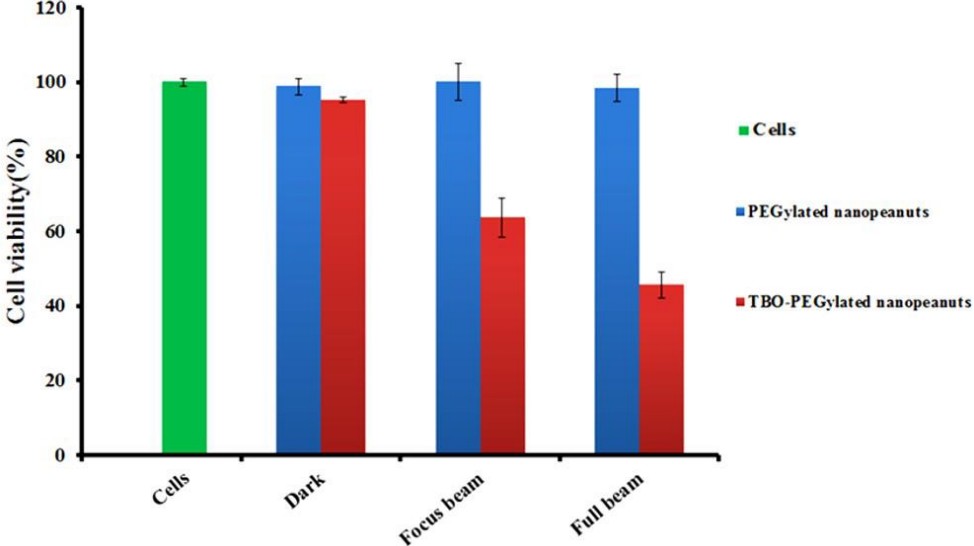

**Figure 6.** Cell viability studies of A549 cells from PEGylated and TBO-PEGylated nanopeanuts (20 μg/mL) in the MTT assay. A549 cells were incubated with nanopeanuts for 8 h at 37 °C. After removal of uninternalized nanopeanuts by PBS wash, A549 cells were subjected to 3-min exposure to a 1230-nm femtosecond Cr: forsterite laser, followed by an additional 16 h incubation. The photodynamic efficacy was evaluated using two different beam size conditions: Focused and collimated beam. The collimated beam has 104 mW/cm$^2$ in power density with a beam diameter of 3–4 mm. The focused beam has $5.7 \times 10^5$ W/cm$^2$ with a beam diameter of 1.5 μm, estimated based on Radius = 0.61 × (λ/NA). The cells -alone group were also subjected to laser irradiation. The dark group represents A549 cells treated with no laser irradiation.

We also examined the in vivo distribution of interstitially administered TBO-PEGylated nanopeanuts in the mouse ear pinna using the same imaging system. An ICR mouse was anesthetized, and the nanopeanuts were subcutaneously injected into the ear pinna. Then, the ear was mounted on the stage of the microscope for further observation. In a typical image, the TPF (Figure 9a) and THG (Figure 9b) contrasts revealed the distribution of nanopeanuts (indicated by white arrows and the area enclosed by white dashed contour). Strong THG contrast in capillaries (outlined by the yellow dashed line in Figure 9b) originates from the flowing red blood cells. These nanopeanuts were drained by a lymphatic vessel (outlined by the yellow dashed line in Figure 9c) near the capillary. These data showed that the pharmacokinetics and efficacy of TBO-PEGylated nanopeanuts can be tracked in vivo in microenvironments, which is a potential theranostic agent for photodynamic therapy in deep tissues.

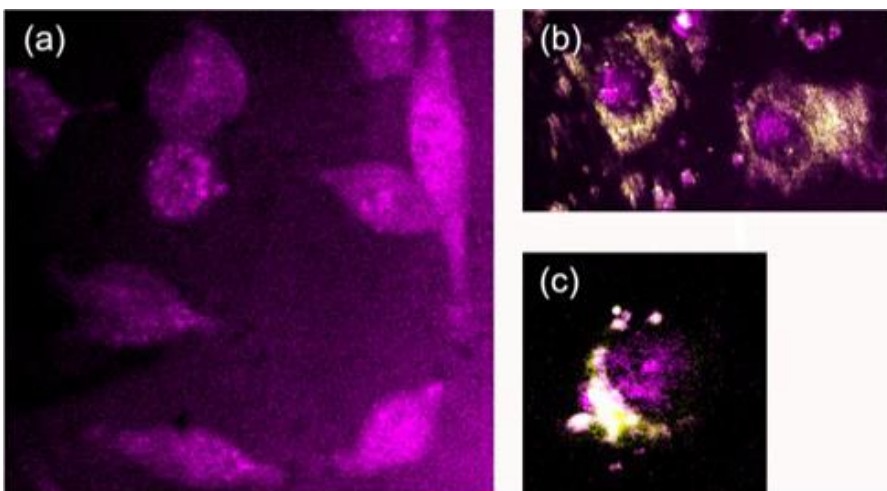

**Figure 7.** (**a**) Microscopy image of A549 lung cancer cells alone. Cells treated with (**b**) PEGylated nanopeanuts and (**c**) TBO-PEGylated nanopeanuts show strong TPF contrast. The magenta color and yellow color represent THG and TPF, respectively. Fields of view: (**a**) 120 μm × 120 μm; (**b**) 90 μm × 50 μm; (**c**) 40 μm × 40 μm.

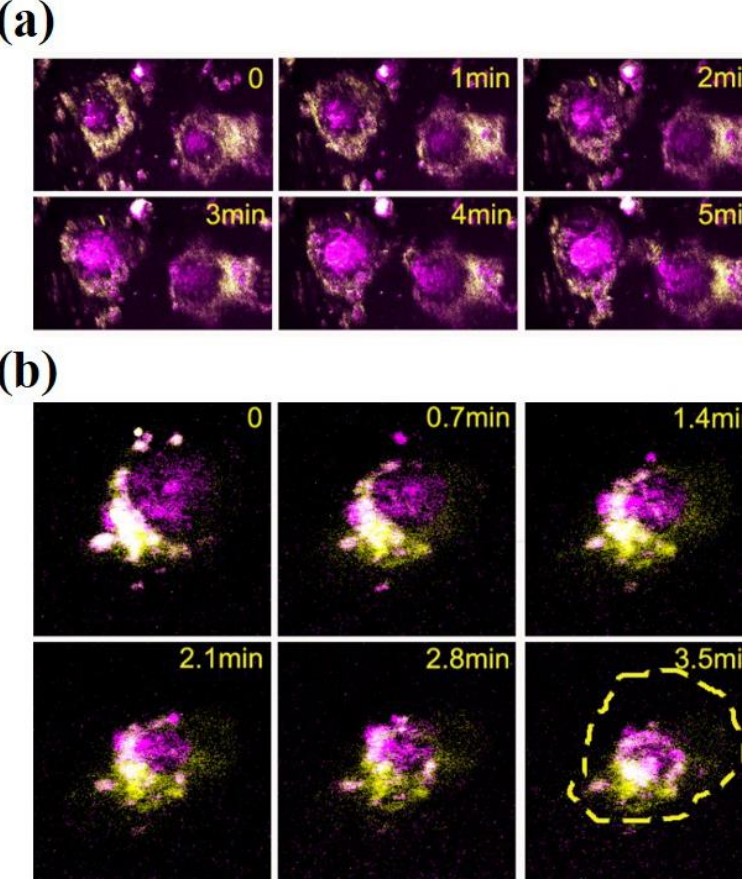

**Figure 8.** Dynamic THG (magenta color) and TPF (yellow color) images of A549 lung cancer cells treated with (**a**) PEGylated nanopeanuts and (**b**) TBO-PEGylated nanopeanuts. After 5 min of illumination with a femtosecond Cr: forsterite laser at a power density of $6.5 \times 10^6$ W/cm$^2$, cells treated with PEGylated nanopeanuts were not affected, while cells treated with TBO-PEGylated nanopeanuts apparently shrank after 3 min of exposure (yellow dashed contour indicates the original size of cells). Fields of view: (**a**) 90 μm × 50 μm; (**b**) 40 μm × 40 μm.

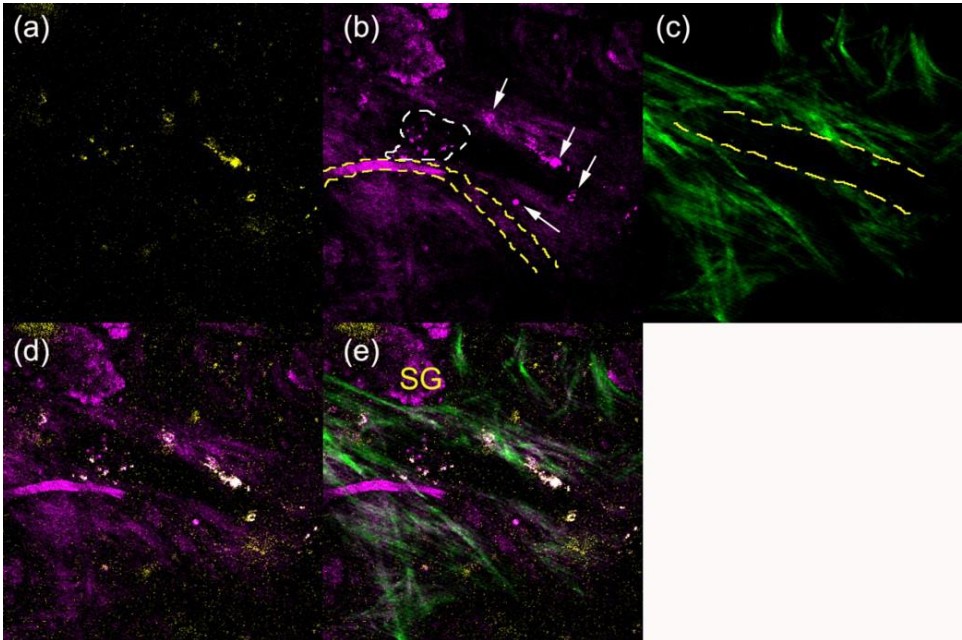

**Figure 9.** In vivo (**a**) TPF (yellow color), (**b**) THG (magenta), (**c**) second harmonic generation (green color), and (**d**,**e**) combined microscopy images of TBO-PEGylated nanopeanuts in the ears of ICR mice. SG: Sebaceous gland. White arrows and the area enclosed by the white dashed contour in (**b**) indicate the positions of nanopeanuts. Yellow dashed lines in (**b**) and (**c**) outline the capillary and lymphatic vessels, respectively.

## 4. Discussion

We have demonstrated a new NIR-active photodynamic platform by plasmon resonant-enhanced two-photon luminescence and singlet oxygen generation from conjugated PS. The NIR surface plasmon bands that appeared beyond 1000 nm allowed us to excite the unique nanopeanut structure in the second biological transparency window. Two-photon excited electrons in TBO may either radiate fluorescence or be coupled to the non-radiative triplet states through intersystem crossing (ISC). The dark state triplet electrons could further generate single oxygens in solutions. In addition, the enhanced fluorescence from TBO-conjugated nanopeanuts, giving stronger contrast than nanopeanuts alone, makes TBO-conjugated nanopeanuts more promising theranostic agents in deep tissues. Inevitably, direct conjugation of TBO to the gold surface will bring severe quenching even though it could still emit substantial fluorescence for two-photon imaging. The reason TBO fluorescence was not completely quenched by gold nanopeanuts could be due to the residual CTAB surfactant on the surface of gold [35]. Many other works that have also employed TBO conjugation have similar issues [36,37]. Thin-layer polyelectrolyte deposition could mitigate this problem [37]. Following this surface modification strategy, we could improve the two-photon fluorescence yields of TBO-PEGylated nanopeanuts in the future.

**Supplementary Materials:** The following are available online at https://www.mdpi.com/article/10.3390/app112210875/s1. Figure S1: HRTEM image and EDX analysis of Au nanopeanuts. Spectra 1–5 represent different sites analyzed in the Ag/Au nanoshell. Au nanopeanuts were prepared by adding 700 µL of $HAuCl_4$ into Au NR@Ag colloidal solution containing CTAB and AA., Figure S2: (a) TEM images and (b) UV-Vis-NIR spectra of Au nanopeanuts obtained from the addition of different volumes (140, 280, 420, and 700 µL) of $HAuCl_4$ into Au NR@Ag colloidal solutions containing CTAB and AA. Insets in each figure present the corresponding colloidal color and the amplified TEM of the single nanopeanut. Figure S3: HRTEM images and EDX analysis of Au nanopeanuts obtained by adding different volumes (140, 280, and 420 µL) of $HAuCl_4$ into Au NR@Ag colloidal solutions containing CTAB and AA. Spectra 1~5 shown for each nanopeanut represent different spot sites

analyzed in the Ag/Au shell. Figure S4: UV-Vis spectra taken as a function of TBO concentration. The inset shows the standard linear calibration curve of TBO. Figure S5: Fluorescence intensity of Singlet Oxygen Sensor Green mixed with TBO-PEGylated nanopeanuts (20 µg/mL) exposed to a 633-nm (200 mW/cm$^2$) diode laser for different irradiation times. Figure S6: Fluorescence intensity of TBO-PEGylated nanopeanuts and free TBO measured from a conventional HORIBA Fluoromax-4 spectrometer by the selection of excitation wavelength 580 nm of xenon arc-lamp. The TBO concentration was fixed at 0.23 µM for both TBO-PEGylated nanopeanuts and free TBO molecules. Figure S7: Fluorescence intensity of Singlet Oxygen Sensor Green mixed with nanopeanuts (20 µg/mL) exposed to a 1230-nm femtosecond Cr: forsterite laser (104 mW/cm$^2$) for 3 min.

**Author Contributions:** Data processing and multiphoton imaging, L.-Z.G.; animal handling and multiphoton imaging, C.-H.W.; synthesis of Au nanopeanuts and material characterization, M.-F.T., F.-Y.C. and V.S.; cell culture and ample preparation, Z.-U.T.; experimental design and sample preparation, Z.Z. and T.-M.L.; material design, C.-S.Y. All authors have read and agreed to the published version of the manuscript.

**Funding:** This work was funded by the National Science Council, grant numbers NSC 100-2628-E-002-006, NSC 100-2627-M-006-007, and NSC 100-2119-M-006-021, the National Health Research Institute of Taiwan, grant number NHRI-EX101-9936EI, and The Science and Technology Development Fund, Macau SAR, File no. 122/2016/A3, 018/2017/A1, 0011/2019/AKP, 0120/2020/A3, and 0026/2021/A.

**Institutional Review Board Statement:** The study was conducted according to the institution's guidelines for the maintenance and use of laboratory animals in research. All of the experimental protocols involving live animals were reviewed and approved by the Animal Experimentation Committee of the National Taiwan University Hospital. All mice were acclimatized in plastic cages and cared for by the animal center.

**Acknowledgments:** This work was supported by the National Science Council (NSC 100-2628-E-002-006, NSC 100-2627-M-006-007 and NSC 100-2119-M-006-021), the National Health Research Institute (NHRI-EX101-9936EI) of Taiwan, and The Science and Technology Development Fund, Macau SAR (File no. 122/2016/A3, 018/2017/A1, FDCT/0011/2019/AKP, 0120/2020/A3, and 0026/2021/A).

**Conflicts of Interest:** The authors declare no conflict of interest.

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
