# Peer review of "Plasmon Resonant Two-Photon Luminescence Inducing Photosensitization and Nonlinear Optical Microscopy In Vivo by Near-Infrared Excitation of Au Nanopeanuts"

_applsci, doi:10.3390/app112210875_

Round 1

Reviewer 1 Report

The manuscript cannot be accepted for publication. There is no significant novelty in this manuscript to justify a publication.

The manuscript is in some part confused and important results were not well measured by the authors. About the absorption, some of the measurements were done in a high concentration, or some problem related to scattering could exist. Scatter induced by nanoparticles aggregation in solution. For example, Figures 2 (c) and (d) present somehow a saturation in the absorption between 570 nm up to 900 nm.

Obs.: absorbance does not have units, even a.u.

There were no one-photon induced emission spectra of the nanoparticles.

About the emission observed at Figure 5, I think that the enhancement of the emission is occurring because of the optical filter, let’s say, distinct absorption at the emission region. As can be seen in Figure 2 there is a high absorbance between 550 nm and 900 nm. The authors must provide a way to prove that this little change in emission is really a Surface-Plasmon-resonance.

In the way that the manuscript is present and some results that must be proved, the manuscript cannot be accepted. Besides, the visual presentation of the manuscript must be improved a lot. Figures were in low resolution and confusing, not clear at all.

Reviewer 2 Report

Title: Plasmon resonant two-photon luminescence inducing photosensitization

and nonlinear optical microscopy in vivo by near-infrared excitation of Au

nanopeanuts

Journal: Applied Sciences

Review comments

This manuscript reported a research on photodynamic therapy (PDT) using a photosensitizer (TBO) conjugated nanopeanuts that enabled two photon induced generation of singlet oxygen and two photon luminescence when 1230 nm femotosecond pulse laser was used as a light source. Authors provided comparison study of A549 cell viability, showing the efficacy of more than 50 % for suppressing the viability with TBO conjugated nanopeanuts. I believe this manuscript is worth being published but only after some points of concern are addressed carefully as follows:

  1. Authors need to provide clearer picture of generating oxygen singlelet by TBO in the manuscript. When TBO conjugated nanopeanuts were exposed by the femtosecond pulsed light at 1230 nm, localized plasmon occurred at 1230 and its increased local fields allowed two photon absorption more effectively. Then TBO could nonlinearly absorb light at half the wavelength of the light source and emitted fluorescence. In this case, how could such fluorescence generate oxygen singlet? In contrast, what about the electrons excited by two photon absorption for generation of oxygen singlet without fluorescence as a non-radiative process? I guess these two processes could occur.
  2. Authors need to offer more details of the pulsed laser source such as full width at half maximum value of the pulsed laser in wavelength domain and the manufacturer with its part number. This spectral range may be important to determine the borderline of the photon energies into which two photon absorption can excite, compared to the localized surface plasmon bands.
  3. What is the pulse peak power of the laser used in experiment? How different is this from those presented in the manuscript? This pulse peak power could be appreciated as a level for non-damaging of cells as presented in Fig. 6.
  4. What is the penetration depth when this laser is used at 1230 nm?
  5. Line 396: what do you mean by this sentence? (enhancement could be 100)
  6. What are the excitation and emission wavelengths of TBO?
  7. What is the factor of the local field enhancement induced by Au nanopeanut plamsons?
  8. Does “full beam” mean “collimated beam” in Figure 6?
  9. Lines 70-72. These lines are difficult to understand. Did this fluorescence emit from nanopeanuts or TBO? If it came from TBO, the distance between TBO and nanopeanut surface could determine the fluorescence enhancement or quenching. If it was less than 10 nm, fluorescence would quench due to non-radiative transfer of excited dipoles directly to surface plasmons at emission wavelength. Authors need to give clearer explanation to address this point.

Round 2

Reviewer 1 Report

The manuscript has been improved and now can be accepted for publication.

Just some corrections for the main manuscript and SI, put only Absorbance in the Figures related, it does not need to put O.D., O. D. is the same as absorbance. At the intensity graphics, use counts instead of count.

A question to be checked in the sentence: "The penetration depth at 1230 nm is about 425 μm. Our home-build laser has a 25 nm full-width at half maximum in the power spectrum, 100 fs pulse-width, 110 MHz pulse repetition rate. The average power after the objective was 60 mW, corresponding to a 5454 W peak power for each laser pulse."

These numbers are correct? 110MHz, which means a 9 ns between pulses?

5454 W peak power? just double-check it.

Reviewer 2 Report

It has been improved for point of concerns for publication 

Author Response

Thanks for reviewer's agree for publication.

Sincerely,

Tzu-Ming Liu